# HEADMAP: LOCATING AND ENHANCING KNOWLEDGE CIRCUITS IN LLMS

**Xuehao Wang**[1,2,*]**, Liyuan Wang**[2]**, Binghuai Lin**[2]**, Yu Zhang**[1,†]
[1]Southern University of Science and Technology
[2]Tencent Technology Co., Ltd, China
{xuehaowangfi,summersunshine366,linbinghuai,yu.zhang.ust}@gmail.com

## ABSTRACT

Large language models (LLMs), through pretraining on extensive corpora, encompass rich semantic knowledge and exhibit the potential for efficient adaptation to diverse downstream tasks. However, the intrinsic mechanisms underlying LLMs remain unexplored, limiting the efficacy of applying these models to downstream tasks. In this paper, we explore the intrinsic mechanisms of LLMs from the perspective of knowledge circuits. Specifically, considering layer dependencies, we propose a layer-conditioned locating algorithm to identify a series of attention heads, which is a knowledge circuit of some tasks. Experiments demonstrate that simply masking a small portion of attention heads in the knowledge circuit can significantly reduce the model's ability to make correct predictions. This suggests that the knowledge flow within the knowledge circuit plays a critical role when the model makes a correct prediction. Inspired by this observation, we propose a novel parameter-efficient fine-tuning method called HeadMap, which maps the activations of these critical heads in the located knowledge circuit to the residual stream by two linear layers, thus enhancing knowledge flow from the knowledge circuit in the residual stream. Extensive experiments conducted on diverse datasets demonstrate the efficiency and efficacy of the proposed method. Our code is available at https://github.com/XuehaoWangFi/HeadMap.

## 1 INTRODUCTION

Large language models (LLMs) have demonstrated strong language understanding capabilities (Radford et al., 2018; 2019; Kenton & Toutanova, 2019; Lewis et al., 2020; Brown et al., 2020) by pertaining on large corpora and can achieve excellent performance on downstream tasks with fine-tuning (Houlsby et al., 2019; Li & Liang, 2021; Hu et al., 2021; Chen et al., 2024; Wei et al., 2024; Yu et al., 2024). Consequently, LLMs are applied to a wide range of fields, which creates a growing need to understand model behavior. However, due to the complex nonlinear interactions within these models, understanding their intrinsic mechanisms remains a significant challenge.

To understand the intrinsic mechanisms of models, previous works (Meng et al., 2022; Wang et al., 2022; Zhang et al., 2024) analyze the roles of different components (e.g., Multi-Head Attention (MHA) blocks and Feed-Forward Network (FFN) blocks) using the causal mediation analysis (CMA) method (PEARL, 2001; Vig et al., 2020) on well-constructed sentences. In particular, Zhang et al. (2024) proposes precise fine-tuning of key attention heads based on their findings to improve the performance of LLMs on mathematical computation tasks. While effective, these studies focus on specific tasks and sentence formats, limiting their scalability.

To address this limitation, we analyze the mechanisms of LLMs from the perspective of multi-head attention. Through experiments, we find that masking a single attention head, randomly masking different heads across various layers, or employing simple greedy masking does not significantly impact the model's ability to make correct predictions. This indicates the presence of a complex structure in LLMs that affects the model's ability to make accurate predictions, and that the behavior

---

* Work done during Xuehao Wang was visiting Tencent as a research intern.
† Corresponding author.

of attention heads is influenced by those in earlier layers. Based on these findings, we propose a layer-conditioned locating algorithm, which accounts for the dependency relationships between different layers, to identify important attention heads layer by layer. Using this method, we can locate a series of attention heads crucial for the ability to make correct predictions on a specific task. We refer to this series of attention heads across all layers as the knowledge circuit (Yao et al., 2024).

In addition to enhancing the understanding of LLMs behavior, since the forward pass of LLMs can be considered as a residual stream where each component of LLMs reads inputs from it and writes outputs to it (Wang et al., 2022; Todd et al., 2024; Zhang et al., 2024), the knowledge circuit identified by the proposed layer-conditional locating algorithm could help improve the performance of LLMs on downstream tasks by enhancing the influence of the knowledge flow in it. To achieve that, we propose a novel parameter-efficient fine-tuning (PEFT) method called HeadMap. Instead of updating all the attention heads as did in conventional PEFT methods, the proposed HeadMap method only transforms the output of attention heads in the identified knowledge circuit via up and down projection and adds the transformed output into the residual stream,helping LLMs better adapt to downstream tasks. The number of learnable parameters in HeadMap is small, since the number of attention heads in the knowledge circuit is not large, leading to its parameter efficiency. Through extensive evaluations on diverse datasets, we demonstrate that the proposed HeadMap method achieves comparable performance to the baseline with fewer parameters. Additionally, the HeadMap method is complementary to other PEFT methods (e.g., LoRA (Hu et al., 2021)), and combining them could significantly enhance the performance.

The contributions of this paper include the following points: (i) We propose the layer-conditioned locating algorithm to identify the existence of knowledge circuits in LLMs that significantly influence the ability to make correct predictions; (ii) Based on knowledge circuits, we proposed the HeadMap method to enhance the knowledge circuits in the residual stream during the forward pass of LLMs, aiding the model in better adapting to downstream tasks; (iii) Experiments on extensive datasets demonstrate the parameter efficiency and effectiveness of the proposed HeadMap method.

## 2 RELATED WORKS

**Mechanistic Interpretability of Large Language Models.** As LLMs play increasingly important roles across various domains (Obermeyer et al., 2019; Rudin, 2019; Bender et al., 2021), understanding the underlying mechanisms behind their varying performance on different tasks has become a growing research focus (Vig et al., 2020; Meng et al., 2022; Wang et al., 2022; Todd et al., 2024; Zhang et al., 2024). To investigate the underlying mechanisms of LLMs, Vig et al. (2020) employ causal mediation analysis (CMA) (PEARL, 2001) to interpret the instinct mechanism of LLM. Following (Vig et al., 2020), Meng et al. (2022) apply CMA method to identify the crucial activations for the model's factual predictions, while Todd et al. (2024) employ CMA in the In-Context Learning (ICL) scenario and reveal that some attention heads in LLMs transmit compact representations corresponding to the task. To enhance the analysis method proposed in (Vig et al., 2020), Wang et al. (2022) propose the path patching method to analyze the role of attention heads on the indirect object identification task. Inspired by (Vig et al., 2020), Zhang et al. (2024) construct a dataset for mathematical computation tasks and employ a path patching method to identify that only a small subset of attention heads plays a crucial role in this task. Yao et al. (2024) focus on factual recall tasks and ablate the special edge in the computation graph of the language model to identify the knowledge circuits. Although those studies analyze the intrinsic mechanisms of LLMs from various perspectives, their analytical methods heavily rely on well-constructed data and are limited to specific tasks, restricting their application to broader scenarios. In contrast, the proposed layer-conditioned locating algorithm, a simple yet efficient method, can identify the knowledge circuit that plays a crucial role in making correct predictions for various tasks.

**Parameter-Efficient Fine-Tuning.** Due to the increasing size of LLMs, fully fine-tuning them for each downstream task has become increasingly difficult. To address this challenge, various PEFT methods have been proposed. Existing parameter-efficient methods are typically categorized into adapter-based methods, prompt tuning methods, and low-rank adaptation methods. Specifically, adapter-based methods (Houlsby et al., 2019; Mahabadi et al., 2021; Karimi Mahabadi et al., 2021) insert compact modules between transformer layers. Prompt tuning methods (Li & Liang, 2021; Lester et al., 2021; Razdaibiedina et al., 2023; Chen et al., 2024; Jiang et al., 2023) add trainable

tokens as prefixes to input or intermediate sequences. Low-rank adaptation methods (Hu et al., 2021; Kopiczko et al., 2023; Wu et al., 2023; Ding et al., 2023; Valipour et al., 2023; Li et al., 2024; Liu et al., 2024; Zhuang et al., 2024) introduces trainable low-rank matrices to approximate weight updates. For example, LoRA represents the weight update matrix as the product of two low-rank matrices and DoRA (Liu et al., 2024) decomposes pre-trained weights into magnitude and direction while fine-tuning direction by LoRA. Different from those methods, the proposed HeadMap method is built on the critical role of knowledge circuits in LLMs and it only makes updates to the attention heads in the knowledge circuits.

## 3 PRELIMINARY

**Transformers.** In this work, we mainly focus on the autoregressive transformer model, which is adopted by most LLMs. Given a sequence of input token embeddings $\mathbf{X}^0 = [\mathbf{x}_1^0, \mathbf{x}_2^0, \cdots, \mathbf{x}_N^0] \in \mathbb{R}^{N \times d}$, where $N$ is the number of tokens in the input and $d$ is the dimension of embeddings. For the model with $L$ transformer layers, each layer consists of an MHA block and an FFN block. Formally, the hidden state $\mathbf{x}_i^l$ of $i$-th token at layer $l$ is calculated as

$$\mathbf{x}_i^l = \mathbf{x}_i^{l-1} + \mathbf{a}_i^l + \mathbf{m}_i^l, \tag{1}$$

where $\mathbf{a}_i^l$ and $\mathbf{m}_i^l$ denote the output of the MHA block and the FFN block for the $i$-th token at layer $l$, respectively. The calculation of $\mathbf{a}_i^l$ and $\mathbf{m}_i^l$ are introduced in the following.

Each MHA block consists of $H$ attention heads, which can allow for attending to different representation subspaces at different positions in the sequence (Vaswani, 2017). For an individual head $h$ in layer $l$, it can be parameterized by three matrices: $\mathbf{W}_Q^{l,h}, \mathbf{W}_K^{l,h}, \mathbf{W}_V^{l,h} \in \mathbb{R}^{d \times \frac{d}{H}}$. Formally, for the input $\mathbf{X}^{l-1} = [\mathbf{x}_1^{l-1}, \mathbf{x}_2^{l-1}, \ldots, \mathbf{x}_N^{l-1}]$ of layer $l$, the output for attention head $h$ is calculated as:

$$\mathbf{h}^{l,h} = f(\mathbf{X}^{l-1}\mathbf{W}_Q^{l,h}, \mathbf{X}^{l-1}\mathbf{W}_K^{l,h}, \mathbf{X}^{l-1}\mathbf{W}_V^{l,h}), \tag{2}$$

where $f(\mathbf{Q}, \mathbf{K}, \mathbf{V}) = \sigma\left(\frac{\mathbf{Q}\mathbf{K}^T}{\sqrt{d/H}}\right)\mathbf{V}$ and $\sigma$ denotes the softmax function. Thus, the output $\mathbf{A}^l = [\mathbf{a}_1^l, \mathbf{a}_2^l, \cdots, \mathbf{a}_N^l]$ of the MHA block is formulated as

$$\mathbf{A}^l = [\mathbf{h}^{l,1}, \mathbf{h}^{l,2}, \cdots, \mathbf{h}^{l,H}]\mathbf{W}_O^l, \tag{3}$$

where $\mathbf{W}_O^l \in \mathbb{R}^{d \times d}$ is a learnable output matrix.

A FFN block consists of a up projection matrix $\mathbf{W}_{up}^l \in \mathbb{R}^{d \times d'}$ and a down projection matrix $\mathbf{W}_{down}^l \in \mathbb{R}^{d' \times d}$, where $d' > d$. For token $i$, the output of the FFN block at layer $l$ is:

$$\mathbf{m}_i^l = \phi((\mathbf{x}_i^{l-1} + \mathbf{a}_i^l)\mathbf{W}_{up}^l)\mathbf{W}_{down}^l, \tag{4}$$

where $\phi(\cdot)$ is an activation function.

**Next token prediction loss.** Given an LLM $\mathcal{M}$, the input sequence $\mathbf{X}$, and the target sequence $\mathbf{Y} = [\mathbf{y}_1, \mathbf{y}_2, ..., \mathbf{y}_{N_y}]$ with $N_y$ tokens, the next token prediction loss is defined as

$$\mathcal{L}(\mathcal{M}, \mathbf{X}, \mathbf{Y}) = -\frac{1}{N_y}\sum_{i=1}^{N_y} \log \mathcal{M}(\mathbf{y}_i | \mathbf{X}, \mathbf{y}_{<i}). \tag{5}$$

## 4 FROM MULTI-HEAD ATTENTION TO KNOWLEDGE CIRCUIT

It is generally accepted that different attention heads in multi-head attention modules focus on different parts of the input and different subspaces of feature representations (Vaswani, 2017). Therefore, previous works (Todd et al., 2024; Jiang et al., 2024) suggest that a small number of attention heads in LLMs can comprehend the reasoning required by tasks and guide the model to make appropriate predictions for diverse ICL tasks. However, it remains unclear whether this phenomenon persists in more challenging tasks, e.g., commonsense reasoning tasks. In this section, we empirically validate this phenomenon on commonsense reasoning tasks and present some empirical findings.

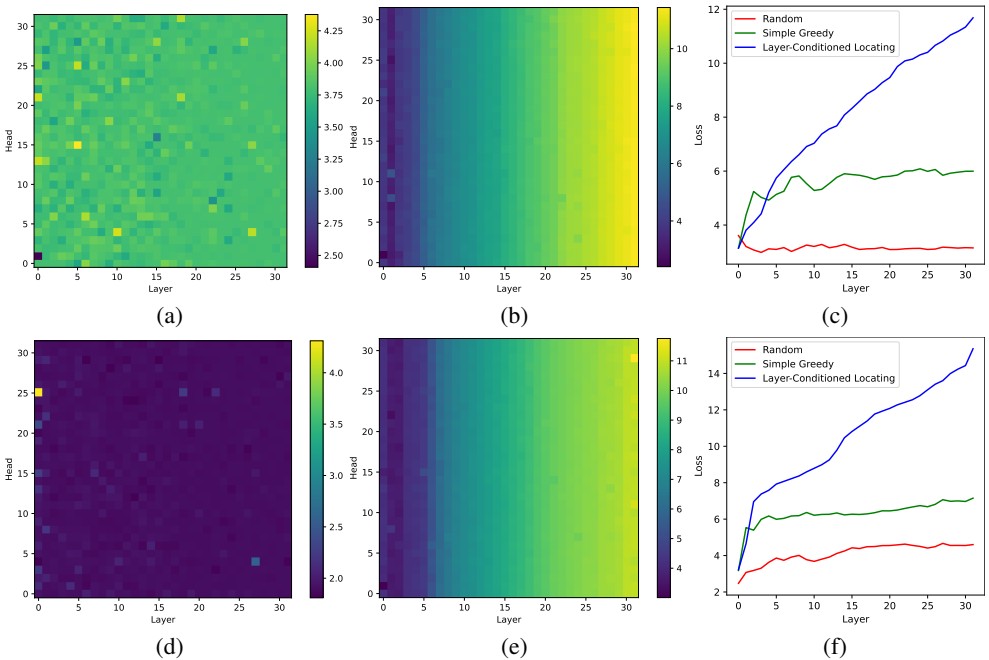

Figure 1: Figures (a) and (d) show the average loss on 64 samples from the SIQA and HellaSwag datasets, respectively, when masking a single attention head. Figures (b) and (e) show the average loss on 64 samples from the SIQA and HellaSwag datasets, respectively, when applying the proposed layer-conditioned locating algorithm. Figures (c) and (f) show the average loss w.r.t. different layers of LLMs on the SIQA and HellaSwag datasets, respectively.

**Model.** In this section, empirical studies are conducted on the LLaMA2-7B model (Touvron et al., 2023). LLaMA2-7B utilizes a decoder-only transformer architecture and is trained on a large text corpus, thus it offers competitive performance across various natural language processing tasks.

**Task and Datasets.** We conduct experiments on commonsense reasoning tasks from the popular evaluation framework lm-evaluation-harness Gao et al. (2024) that have garnered widespread attention. Those tasks are designed to challenge models beyond mere pattern recognition, requiring them to apply real-world knowledge to make inferences. Due to the complexity of the task inputs, performing causal inference analysis through input sequence modifications to identify underlying knowledge circuits, as explored in previous works (Wang et al., 2022; Zhang et al., 2024; Meng et al., 2022), presents challenges. The commonsense reasoning tasks contain eight datasets, including six question-answering datasets (i.e., BoolQ (Clark et al., 2019), PIQA (Bisk et al., 2020), SIQA (Sap et al., 2019), ARC-e (Clark et al., 2018), ARC-c (Clark et al., 2018), and OBQA (Mihaylov et al., 2018)), a context completion dataset (i.e., HellaSwag (Zellers et al., 2019)), and a fill-in-the-blank dataset (i.e., WinoGrande (Sakaguchi et al., 2021)). Following (Todd et al., 2024; Jiang et al., 2024), our analysis focuses on samples that LLMs can predict successfully and utilizes such 64 samples with the lowest next token prediction loss on each dataset to conduct experiments.

## 4.1 EXPERIMENT 1: LOCATING CRITICAL ATTENTION HEADS

Critical attention heads are considered as those who change their outputs and then have a substantial influence on the LLMs' ability to make correct predictions for a certain task. Therefore, a straightforward method to locate critical attention heads is to mask their outputs and observe whether it causes the LLM to make incorrect predictions. That is, to evaluate the importance of attention head $h$ in layer $l$, we set $\mathbf{h}^{l,h} = 0$ and calculate the training loss (i.e. next token prediction loss). If this loss is much larger than the loss without the masking, this attention head may be critical.

**Results.** Based on this simple idea, we conduct experiments on commonsense reasoning tasks. Figure 1(a) and 1(d) illustrate the effect of masking each head in each layer for LLaMA2-7B on SIQA and HellaSwag datasets. The overall results are shown in Figure 4 of Appendix A.1. According to the results, we can see that masking a single attention head does not cause significant changes to

---

**Algorithm 1** Layer-Conditioned Locating Algorithm

---

**Require:** Number of selected attention heads $n$ of each layer, set of selected samples $\mathbb{D}$, model $\mathcal{M}$ with $L$ layers and $H$ heads at each layer.

1: Initialize $\mathbb{H} \leftarrow \emptyset$
2: **for** $l \leftarrow 1$ **to** $L$ **do**
3:      **for** $h \leftarrow 1$ **to** $H$ **do**
4:          $\mathbb{H} \leftarrow \mathbb{H} \cup \{(l, h)\}$;
5:          $loss_{l,h} \leftarrow 0$
6:          **for** $(\mathbf{X}, \mathbf{Y}) \in \mathbb{D}$ **do**
7:              $loss_{l,h} \leftarrow loss_{l,h} + \mathcal{L}(\mathcal{M}_{\mathbb{H}}, \mathbf{X}, \mathbf{Y})$   ▷ Calculate the loss when mask the heads in $\mathbb{H}$
8:          **end for**
9:          $\mathbb{H} \leftarrow \mathbb{H} \setminus \{(l, h)\}$
10:      **end for**
11:      **for** $h \in \arg\_\text{top}\_\text{K}(loss_l, n)$ **do**
12:          $\mathbb{H} \leftarrow \mathbb{H} \cup \{(l, h)\}$                    ▷ Select heads with higher influence of loss
13:      **end for**
14: **end for**
15: **return** $\mathbb{H}$

---

the loss of LLMs. For example, the highest average loss change, the difference between the average loss after and before masking on the SIQA and HellaSwag datasets is only $+3.02$ and $+2.58$, respectively. This suggests that for complex tasks, the loss of LLMs is not driven by the output of a single attention head. Inspired by (Elhage et al., 2021), the knowledge flow in the LLMs can be considered as a residual stream, while MHA and FFN blocks can read and write knowledge in this stream. Therefore, we hypothesize that within LLMs, there exists a knowledge circuit composed of multiple attention heads across different layers, which aids the model in making correct predictions for specific tasks by reading and writing the important knowledge in the residual stream.

## 4.2 EXPERIMENT 2: SIMPLE GREEDY LOCATING ALGORITHM

To have a better understanding of the knowledge circuit, we consider the forward pass of the LLM as a computation graph (Wang et al., 2022; Zhang et al., 2024). Each node in this computation graph denotes a computation component of LLM (i.e., attention heads and FFN blocks) and the edge between different nodes denotes the hidden state, which is calculated from the starting node and is the input of the ending node. Therefore, the knowledge circuit can be viewed as a subgraph of the computational graph in LLMs and facilitates the critical knowledge flow from input to output.

Based on this perspective, a natural approach to locating a knowledge circuit is to make a simple greedy locating. This greedy locating method selects the top-$n$ most critical attention heads in each layer, where the importance of each attention head is calculated as in Section 4.1, and constructs the knowledge circuit as the subgraph of the computation graph consisting of selected attention heads.

**Results.** Here we set $n$ to be 8 and hence the knowledge circuit consists of $8 \times 32$ attention heads. The green lines in Figures 1(c) and 1(f) show the loss change curve when attention heads are masked layer by layer using the aforementioned greedy locating method. The overall results are shown in Figure 6 of Appendix A.3. It is observed that while the initial layers show significant impact, masking attention heads after the 5-th layer does not lead to substantial changes. This contradicts previous findings (Jin et al., 2024), which finds that attention heads in the later layers still play a significant role. A plausible explanation is that this greedy locating method overlooks the inter-layer dependencies within LLMs. Specifically, when critical attention heads in earlier layers are masked, the attention heads in later layers may behave differently, since some attention heads that seem impactful individually might merely reinforce the knowledge provided by earlier critical attention heads, and their influence disappears once those earlier heads are masked.

## 4.3 EXPERIMENT 3: LAYER-CONDITIONED LOCATING ALGORITHM

To address this issue, we propose a layer-conditioned locating algorithm to locate knowledge circuits by taking layer dependencies into consideration. Specifically, the algorithm performs a greedy

search layer by layer, with the loss calculation for each masking attention head depending on the attention heads masked in the previous layers. The detailed process is shown in Algorithm 1, whose time complexity is $\mathcal{O}(HL)$ with $H$ as the number of attention heads per layer in LLMs and $L$ as the total number of layers. Similar to the greedy locating algorithm, we set $n = 8$ and evaluate the layer-conditioned locating algorithm on LLaMA2-7B.

**Results.** Figures 1(b) and 1(e) show the average loss when masking the corresponding attention heads, with critical attention heads in previous layers already masked. The overall results are shown in Figure 5 of Appendix A.2. Figures 1(c) and 1(f) show the trend of the average loss w.r.t. the number of masked layers for selecting 8 heads per layer randomly by the greedy locating algorithm and by layer-conditioned locating algorithm, respectively. The overall results are shown in Figure 6 of Appendix A.3. It can be seen that the knowledge circuit selected by the layer-conditioned locating algorithm has a greater impact on the training loss compared to that selected randomly or by the greedy locating algorithm.

### 4.4 FINDINGS

Based on the above experimental results and analyses, we have the following findings.

**Finding 1.** Masking some attention heads (i.e. the knowledge circuit) significantly impairs the large language model's ability to make accurate predictions, indicating that the knowledge written and read by the knowledge circuit in the residual stream is crucial for correct predictions.

**Finding 2.** Masking redundant attention heads in some cases reduces loss, suggesting that knowledge written and read by certain redundant attention heads in the residual stream may interfere with the model's accurate predictions.

**Finding 3.** For a given task, a knowledge circuit impacts the ability of LLMs to make correct predictions, and knowledge circuits varies across tasks. This suggests that LLMs use different knowledge and patterns for different tasks, and there is no single knowledge circuit that can handle all tasks.

## 5 METHODOLOGY

**Findings 1** and **3** suggest, there exists a crucial subset of attention heads, referred to as the knowledge circuit, that are essential for making the right predictions on the corresponding task. These knowledge circuits enable the model to make accurate predictions by reading and writing important knowledge in the residual stream of the LLM. However, **Finding 2** suggests the presence of numerous redundant attention heads in the model, whose outputs interfere with the knowledge flow in the knowledge circuit, thus leading to incorrect predictions. Inspired by these findings, we propose a novel parameter-efficient fine-tuning method, HeadMap. This method enhances the knowledge flow from knowledge circuits in the residual stream, aiding large language models in making accurate predictions.

Due to the FFN block reading and writing on both the residual stream and MHA output, we propose two variants of the HeadMap method: the first one enhances the residual stream directly (Section 5.1), and the other enhances in the MHA block (Section 5.2). Specifically, the former bypasses the FFN block and $W_O$, directly writing the knowledge from the knowledge circuit into the residual stream, while the latter writes the knowledge into the MHA outputs, thereby affecting both the residual stream and the operation of the FFN block.

### 5.1 HEADMAP FOR HIDDEN STATES

The forward pass of LLMs can be viewed as a long residual stream, where all attention blocks and FFN blocks read from and write to it. Thus, at the $l$-th layer, to enhance the influence of the knowledge circuit located by the layer-conditioned locating algorithm, we map the outputs of the selected attention heads and add them to the hidden state $\mathbf{X}^l$. Formally, the output of layer $l$ is calculated as

$$\mathbf{X}^l = \mathbf{X}^{l-1} + \mathbf{A}^l + \mathbf{M}^l + \text{Map}([\mathbf{h}^{l,s_1}, ..., \mathbf{h}^{l,s_n}]), \tag{6}$$

where $\mathbf{M}^l = [\mathbf{m}_1^l, \mathbf{m}_2^l, \cdots, \mathbf{m}_N^l]$ denotes the output of the FFN block in layer $l$, $\text{Map}(\cdot)$ denotes a neural network, $\mathbb{H}$ denotes the set containing the selected attention heads in the knowledge circuits,

$n$ is the number of selected attention heads per layer, and $(l, s_i) \in \mathbb{H}, \forall i \in [1, n]$. To reduce the number of trainable parameters in $\text{Map}(\cdot)$, we use a bottleneck architecture where we first down-project the outputs of the selected attention heads into $r$-dimension and then up-project them into $d$. Thus, the output can be formulated as

$$\mathbf{X}^l = \mathbf{X}^{l-1} + \mathbf{A}^l + \mathbf{M}^l + \text{Up}\left(\text{Down}([\mathbf{h}^{l,s_1}, ..., \mathbf{h}^{l,s_n}])\right). \tag{7}$$

### 5.2 HeadMap for Outputs of MHA Blocks

In addition to adding directly to the residual stream, another way to enhance the knowledge circuit is to place HeadMap within the MHA. This involves directly adding the mapped output to the output $\mathbf{A}^l$ of the MHA block at the $l$-th layer, resulting in the following output as

$$\mathbf{A}^l = [\mathbf{h}^{l,1}, \mathbf{h}^{l,2}, \cdots, \mathbf{h}^{l,H}]\mathbf{W}_O^l + \text{Up}\left(\text{Down}([\mathbf{h}^{l,s_1}, ..., \mathbf{h}^{l,s_n}])\right). \tag{8}$$

Since we use two linear mappings here, this method is similar to selecting $\{W_O^{l,h} | (l, h) \in \mathbb{H}\}$ to perform LoRA fine-tuning and does not introduce inference delay.

### 5.3 Implemented detials

**Initialization.** The parameters of down projection layers are initialized as $\mathbf{0}$, while the parameters of up-project layers are initialized by Kaiming uniform. Therefore, in the begining of the training process, the HeadMap method does not change the output of LLMs.

**Parameter complexity.** We provide a parameter complexity analysis of the proposed HeadMap method, where the two variants have the same number of parameters. To fine-tune the large language model with hidden dimension $d$, the proposed HeadMap method selects $n$ attention heads at each layer. Thus, the HeadMap method introduces $\mathcal{O}\left((n \times \frac{d}{H} + 1) \times r + (r + 1) \times d\right)$ trainable parameters each layer, where $r$ is the dimension after down-projection and $r \ll d$.

## 6 Experiment

In this section, we empirically evaluate the proposed HeadMap method.

### 6.1 Experiments on Commonsense Reasoning Tasks

In this section, we conduct experiments on commonsense reasoning tasks.

**Datasets.** As introduced before, the commonsense reasoning tasks comprise eight datasets, including BoolQ, PIQA, SIQA, ARC-e, ARC-c, OBQA, HellaSwag, and WinoGrande. Each dataset contains a training and testing set. We use the accuracy metric to measure the performance.

**Baselines.** LLaMA2-7B (Touvron et al., 2023) and LLaMA3-8B (Meta, 2024) are used to be LLMs. We compare the proposed HeadMap method with LoRA (Hu et al., 2021) and DoRA (Liu et al., 2024). LoRA and DoRA are used on $W_Q$, $W_K$, $W_V$, $W_O$, $W_{up}$, and $W_{down}$ at each layer. Additionally, we explore combining HeadMap with LoRA and DoRA by applying LoRA or DoRA to fine-tune $W_V$, $W_{up}$, and $W_{down}$ while implementing HeadMap.

**Setups.** For commonsense reasoning datasets, we fine-tune and evaluate models on each single dataset. The batch size is set to 16, and the AdamW optimizer is used. We use the linear learning rate scheduler and train all methods for 2 epochs with 100 warmup steps. The learning rate is set to 0.0002 for LoRA and DoRA, while set to 0.001 for HeadMap. We set rank $r = 32$ for all methods and $\alpha = 64$ for LoRA and DoRA. We select 8 attention heads for each MHA block and use 64 samples in each dataset to locate the knowledge circuit.

**Results.** The experimental results on commonsense reasoning datasets are shown in Table 1. It can be observed that by enhancing the impact of critical heads, both variants of the HeadMap method achieve comparable results to the baselines while introducing only one-tenth of the parameters. Additionally, we find that HeadMap and LoRA-based methods (i.e., LoRA and DoRA) are complementary as combining LoRA and DoRA with HeadMap achieves better results with fewer parameters.

Table 1: Accuracy (%) on eight commonsense reasoning datasets using LLaMA2-7B and LLaMA3-8B. The best result in each comparison group is in **bold**. HeadMap$_{MHA}$ denotes the variant of HeadMap for outputs of MHA blocks as introduced in Section 5.2 and HeadMap$_{HS}$ denotes the variant of HeadMap for hidden states in Section 5.1.

| Model | # Params (%) | BoolQ | PIQA | SIQA | HellaSwag | WinoGrande | ARC-e | ARC-c | OBQA | Avg. |
|---|---|---|---|---|---|---|---|---|---|---|
| *LLaMA2-7B* | | | | | | | | | | |
| LoRA | 0.95 | 63.52 | 81.34 | 78.56 | 93.78 | 81.93 | 79.71 | 54.78 | 77.40 | 76.38 |
| DoRA | 0.96 | 64.25 | 83.03 | 78.30 | **93.81** | **82.95** | 79.50 | 55.97 | 78.20 | 77.02 |
| HeadMap$_{MHA}$ | 0.08 | 63.94 | 83.19 | 79.07 | 92.74 | 79.40 | 80.30 | **57.00** | 78.60 | 76.78 |
| + LoRA (V, U, D) | 0.66 | 65.38 | **84.66** | 79.58 | 93.62 | 80.58 | 81.65 | 56.40 | 80.20 | 77.76 |
| + DoRA (V, U, D) | 0.67 | 65.93 | 84.06 | 79.79 | 93.67 | 81.93 | 82.03 | **57.00** | **81.00** | 78.18 |
| HeadMap$_{HS}$ | 0.08 | 63.73 | 83.30 | 79.73 | 92.63 | 78.37 | 80.30 | 55.38 | 78.20 | 76.46 |
| + LoRA (V, U, D) | 0.66 | 66.85 | 84.06 | 79.84 | 93.59 | 82.08 | **82.37** | 56.31 | 80.60 | 78.21 |
| + DoRA (V, U, D) | 0.67 | **67.16** | 84.28 | **80.14** | 93.59 | 82.00 | 81.73 | 56.57 | **81.00** | **78.31** |
| *LLaMA3-8B* | | | | | | | | | | |
| LoRA | 0.80 | 76.30 | 86.07 | 80.60 | 96.01 | 86.82 | 90.70 | 78.67 | 83.40 | 84.82 |
| DoRA | 0.81 | **76.73** | 86.40 | 80.60 | 96.13 | 86.42 | 91.37 | 78.75 | 87.60 | 85.50 |
| HeadMap$_{MHA}$ | 0.07 | 75.29 | 89.17 | 81.47 | 95.64 | 85.24 | 92.00 | 78.75 | 88.40 | 85.74 |
| + LoRA (V, U, D) | 0.60 | 76.30 | **90.48** | 81.27 | 96.16 | 87.13 | 92.05 | 80.46 | 88.00 | 86.48 |
| + DoRA (V, U, D) | 0.61 | 76.67 | 89.83 | 81.68 | 96.35 | **87.92** | 92.13 | 79.69 | 87.80 | 86.51 |
| HeadMap$_{HS}$ | 0.07 | 75.35 | 89.28 | 81.53 | 95.53 | 85.40 | 92.17 | 78.92 | 87.80 | 85.75 |
| + LoRA (V, U, D) | 0.60 | 76.24 | 89.39 | 81.52 | **96.53** | 87.29 | 91.84 | **81.06** | **88.60** | **86.56** |
| + DoRA (V, U, D) | 0.61 | 76.48 | 90.04 | **81.73** | 96.44 | 87.29 | **92.30** | 79.78 | 87.00 | 86.38 |

Table 2: Accuracy (%) on five NLU datasets using LLaMA2-7B and LLaMA3-8B. The best result in each comparison group is in **bold**. HeadMap is for outputs of MHA blocks. HeadMap$_{HS}$ is for hidden states.

| Method | # Params (%) | CoLA | MRPC | RTE | SST-2 | WNLI | Avg. |
|---|---|---|---|---|---|---|---|
| *LLaMA2-7B* | | | | | | | |
| LoRA | 0.95 | 49.20 | 70.34 | 87.00 | 96.10 | **59.15** | 72.36 |
| HeadMap$_{MHA}$ | 0.08 | 52.15 | 69.12 | 82.67 | 95.87 | 54.93 | 70.95 |
| + LoRA (V, U, D) | 0.66 | 59.92 | **82.11** | **87.36** | **96.90** | 56.34 | 76.53 |
| HeadMap$_{HS}$ | 0.08 | 47.63 | 77.70 | 85.92 | 95.53 | 54.93 | 72.34 |
| + LoRA (V, U, D) | 0.66 | **64.41** | 80.39 | **87.36** | 96.44 | **59.15** | **77.55** |
| *LLaMA3-8B* | | | | | | | |
| LoRA | 0.80 | 65.48 | 85.29 | 88.81 | 96.10 | 70.42 | 81.22 |
| HeadMap$_{MHA}$ | 0.07 | 62.75 | 85.78 | 88.09 | 95.76 | 71.83 | 80.84 |
| + LoRA (V, U, D) | 0.60 | 67.82 | 87.25 | **90.25** | 96.33 | **80.28** | **84.39** |
| HeadMap$_{HS}$ | 0.07 | 63.85 | 85.29 | 88.81 | 95.99 | 69.01 | 80.59 |
| + LoRA (V, U, D) | 0.60 | **68.77** | **87.75** | 89.89 | **96.44** | 77.46 | 84.06 |

Moreover, on LLaMA2-7B, combining HeadMap for hidden states with DoRA achieves the best performance on average (+1.29% compared to DoRA) and some tasks (i.e. BoolQ, SIQA, and OBQA), while HeadMap for hidden states with LoRA achieves the best performance on average (+1.06% compared to DoRA) and several tasks (i.e. HellaSwag, Arc-c, and OBQA) on LLaMA3-8B.

## 6.2 EXPERIMENTS ON NATURAL LANGUAGE UNDERSTANDING TASKS

In this section, we conduct experiments on Natural Language Understanding (NLU) tasks.

**Datasets.** For NLU tasks, we compare baselines on five datasets, including CoLA (Warstadt et al., 2018), MRPC (Dolan & Brockett, 2005), RTE (Wang et al., 2019), SST-2 (Socher et al., 2013), and WNLI (Wang et al., 2019). We use Matthew's correlation for CoLA and accuracy for other tasks to measure the performance of each method.

**Baselines.** For NLU tasks, we compare the proposed HeadMap method with LoRA on LLaMA2-7B and LLaMA3-8B. LoRA are used on $W_Q$, $W_K$, $W_V$, $W_O$, $W_{up}$, and $W_{down}$ at each layer. Moreover, we combine HeadMap with LoRA.

**Setups.** Similar to commonsense reasoning datasets, we fine-tune and evaluate models on each single dataset. The batch size is set to 16, and the AdamW optimizer is used. We use the linear learning rate scheduler and train all methods for 2 epochs with 100 warmup steps. For the learning rate of each method, we perform the grid search in $\{0.0001, 0.0002, 0.0004, 0.0008, 0.001\}$ on each dataset. We set rank $r = 32$ for all methods and $\alpha = 64$ for LoRA. We select 8 attention heads for each MHA block. We select 8 attention heads for each MHA block and use 64 samples with the lowest loss in each dataset to locate the knowledge circuit.

Table 3: Accuracy (%) on eight commonsense reasoning datasets using LLaMA2-7B and LLaMA3-8B for different selection strategies. HeadMap is for outputs of MHA blocks. HeadMap$_{HS}$ is for hidden states.

| Method | BoolQ | PIQA | SIQA | HellaSwag | WinoGrande | ARC-e | ARC-c | OBQA | Avg. |
|---|---|---|---|---|---|---|---|---|---|
| *LLaMA2-7B* | | | | | | | | | |
| HeadMap$_{MHA}$ (Random) | 62.63 | 82.48 | 78.25 | 91.88 | 78.14 | 78.07 | 54.78 | 75.00 | 75.15 |
| HeadMap$_{MHA}$ | 63.94 | 83.19 | 79.07 | 92.74 | 79.40 | 80.30 | 57.00 | 78.60 | 76.78 |
| HeadMap$_{HS}$ (Random) | 63.27 | 83.57 | 78.30 | 92.51 | 76.64 | 78.79 | 53.41 | 75.40 | 75.24 |
| HeadMap$_{HS}$ | 63.73 | 83.30 | 79.73 | 92.63 | 78.37 | 80.30 | 55.38 | 78.20 | 76.46 |
| *LLaMA3-8B* | | | | | | | | | |
| HeadMap$_{MHA}$ (Random) | 74.46 | 87.92 | 80.50 | 95.35 | 84.05 | 90.95 | 77.90 | 86.80 | 84.74 |
| HeadMap$_{MHA}$ | 75.29 | 89.17 | 81.47 | 95.64 | 85.24 | 92.00 | 78.75 | 88.40 | 85.74 |
| HeadMap$_{HS}$ (Random) | 74.34 | 88.25 | 80.30 | 95.40 | 83.82 | 91.08 | 77.30 | 86.80 | 84.66 |
| HeadMap$_{HS}$ | 75.35 | 89.28 | 81.53 | 95.53 | 85.40 | 92.17 | 78.92 | 87.80 | 85.75 |

Table 4: Accuracy (%) on eight commonsense reasoning datasets using LLaMA2-7B and LLaMA3-8B for different number of selected heads. HeadMap is for outputs of MHA blocks. HeadMap$_{HS}$ is for hidden states.

| Method | # Params (%) | BoolQ | PIQA | SIQA | HellaSwag | WinoGrande | ARC-e | ARC-c | OBQA | Avg. |
|---|---|---|---|---|---|---|---|---|---|---|
| *LLaMA2-7B* | | | | | | | | | | |
| HeadMap$_{MHA}$ (4) | 0.07 | 64.50 | 82.10 | 79.93 | 92.59 | 78.93 | 79.29 | 55.38 | 78.20 | 76.37 |
| HeadMap$_{MHA}$ (8) | 0.08 | 63.94 | 83.19 | 79.07 | 92.74 | 79.40 | 80.30 | 57.00 | 78.60 | 76.78 |
| HeadMap$_{MHA}$ (16) | 0.10 | 64.28 | 83.41 | 79.32 | 92.83 | 79.48 | 79.50 | 56.06 | 79.00 | 76.74 |
| HeadMap$_{HS}$ (4) | 0.07 | 64.04 | 82.92 | 78.81 | 92.56 | 79.01 | 79.67 | 54.18 | 79.20 | 76.30 |
| HeadMap$_{HS}$ (8) | 0.08 | 63.73 | 83.30 | 79.73 | 92.63 | 78.37 | 80.30 | 55.38 | 78.20 | 76.46 |
| HeadMap$_{HS}$ (16) | 0.10 | 64.28 | 83.68 | 78.97 | 92.80 | 78.85 | 79.63 | 55.72 | 77.60 | 76.44 |
| *LLaMA3-8B* | | | | | | | | | | |
| HeadMap$_{MHA}$ (4) | 0.06 | 75.44 | 88.63 | 81.68 | 95.68 | 84.93 | 92.30 | 79.30 | 87.80 | 85.72 |
| HeadMap$_{MHA}$ (8) | 0.07 | 75.29 | 89.17 | 81.47 | 95.64 | 85.24 | 92.00 | 78.75 | 88.40 | 85.74 |
| HeadMap$_{MHA}$ (16) | 0.08 | 74.31 | 89.23 | 81.83 | 95.86 | 86.11 | 90.91 | 79.10 | 87.00 | 85.54 |
| HeadMap$_{HS}$ (4) | 0.06 | 75.05 | 88.36 | 81.17 | 95.35 | 85.08 | 91.75 | 79.86 | 87.80 | 85.48 |
| HeadMap$_{HS}$ (8) | 0.08 | 75.35 | 89.28 | 81.53 | 95.53 | 85.40 | 92.17 | 78.92 | 87.80 | 85.75 |
| HeadMap$_{HS}$ (16) | 0.08 | 75.08 | 88.74 | 81.63 | 95.76 | 85.63 | 92.13 | 79.44 | 87.60 | 85.75 |

**Results.** The experimental results on diverse NLU datasets are shown in Table 2. As can be seen, HeadMap achieves comparable performance with fewer parameters. Additionally, compared to directly only applying LoRA for fine-tuning, combining it with HeadMap significantly enhances performance. Specifically, HeadMap for hidden states has much better performance on average ($+5.19\%$ compared to LoRA) and on the CoLA, RTE, and SST-2 datasets with LLaMA2-7B as the LLM. For LLaMA3-8B, HeadMap for outputs of MHA blocks achieves the best results on average ($+2.84\%$ compared to LoRA) and on the CoLA, MRPC, and SST-2 datasets.

## 6.3 ABLATION STUDY

**Effect of layer-conditioned locating algorithm.** Here, we conduct ablation study on the proposed layer-conditioned locating algorithm. Following the hyperparameters setup in Section 6.1, we compared it with a random selection algorithm on commonsense reasoning datasets. According to the results shown in Table 3, we can see that attention heads selected by the layer-conditioned locating algorithm outperforms random selection across all datasets and models, demonstrating the effectiveness of the proposed layer-conditioned locating algorithm.

**Sensitivity to the number of selected heads per layer.** Here, we investigate the impact of the number of selected attention heads to the fine-tuning performance with the same settings of other hyperparameters as Section 6.1. According to the results shown in Table 4, we can observe that the performance of the proposed HeadMap method remains stable w.r.t. the number of heads across the two LLMs, which indicates that critical attention heads in each layer are limited and adding more does not lead to performance improvement.

## 6.4 ANALYSIS

**Overlapping of knowledge circuits in different datasets.** In this experiment, we analyze the overlapping of attention heads in knowledge circuits identified in different datasets by adopting the

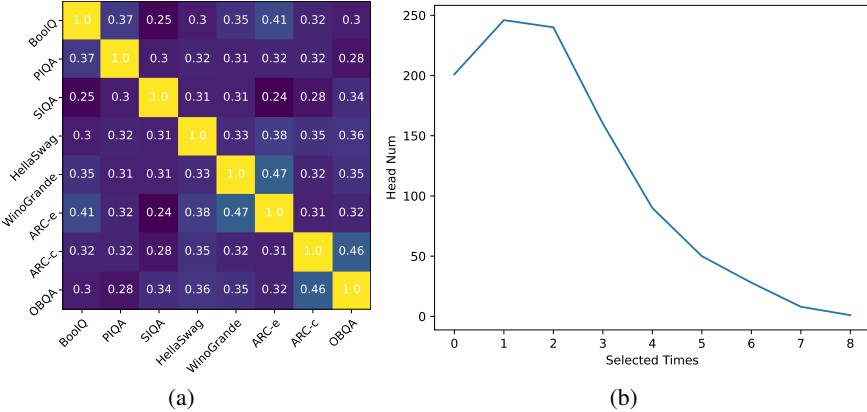

(a)                    (b)

Figure 2: (a) The overlap rate of knowledge circuits between different tasks on LLaMA2-7B. (b) Distribution of attention head selection frequency.

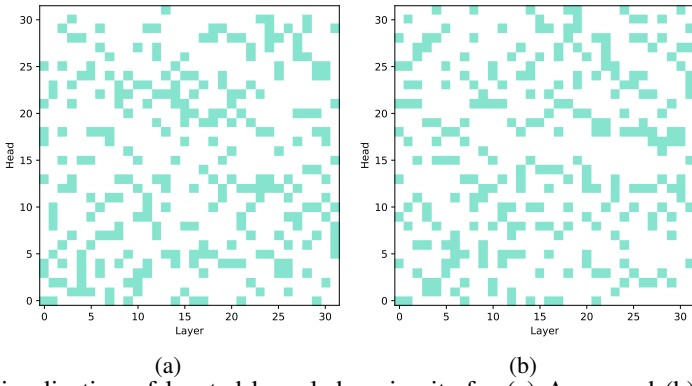

(a)                    (b)

Figure 3: The visualization of located knowledge circuits for (a) Arc-e and (b) WinoGrande on LLaMA2-7B.

same setups as Section 6.1. Specifically, we calculate the overlap rate of knowledge circuits between datasets $a$ and $b$ as $\frac{|\mathbb{H}_a \cap \mathbb{H}_b|}{L \times H}$.

The overlap rate between different tasks on LLaMA2-7B and LLaMA3-8B are shown in Figures 2(a) and 8, respectively. The visualization of knowledge circuits for the Arc-easy and WinoGrande datasets on LLaMA2-7B is shown in Figure 3. It can be observed that for most tasks, the overlap rate of their identified knowledge circuits is low. This suggests that the knowledge circuits tend to vary significantly due to differing knowledge requirements for different tasks. However, some tasks show higher overlap rates. For example, when selecting only 8 attention heads per layer, the overlap rate between the knowledge circuits in the ARC-e and WinoGrande datasets reaches 0.47, reflecting the similar knowledge and behaviors needed for model reasoning in these tasks.

**The number of times for attention heads being selected.** In Figure 2(b), we record the number of times being selected for each attention head over the eight commonsense reasoning datasets. It can be seen that only a small number of attention heads are located by many different tasks, indicating that most attention heads play a crucial role in only a few specific tasks.

## 7    CONCLUSION

In this paper, we propose the layer-conditioned locating algorithm to identify knowledge circuits containing a series of attention heads that play a crucial role in a specific task. After locating the knowledge circuit, we propose the HeadMap method to enhance the influence of the knowledge circuit on the residual stream, allowing the model to better adapt to downstream tasks. Extension experiments demonstrate that HeadMap achieves comparable performance with few parameters. Moreover, HeadMap and LoRA-based methods are complementary, and their combination effectively improves model performance. In our future work, we are interested in applying HeadMap to more applications.

ACKNOWLEDGEMENTS

This work is supported by National Key R&D Program of China 2022ZD0160300 and NSFC key grant under grant no. 62136005.

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

# A ADDITIONAL EXPERIMENTAL RESULTS

## A.1 LOCATING THE CRITICAL ATTENTION HEAD

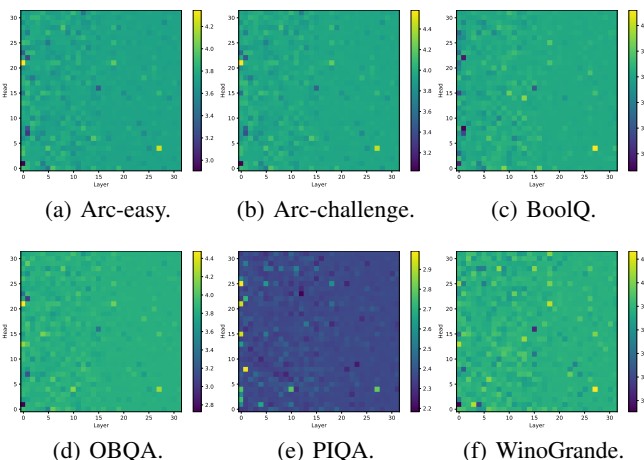

(a) Arc-easy.          (b) Arc-challenge.          (c) BoolQ.

(d) OBQA.          (e) PIQA.          (f) WinoGrande.

Figure 4: When masking a single attention head, the average loss of 64 samples from each dataset.

The average loss in model predictions when a single attention head is masked is shown in Figure 4. It can be observed that masking a single attention head does not have a significant impact across different datasets.

## A.2 LAYER-CONDITIONED LOCATING ALGORITHM

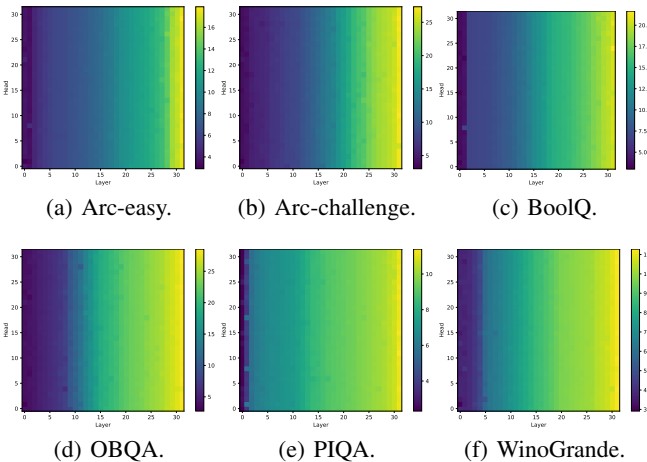

(a) Arc-easy.          (b) Arc-challenge.          (c) BoolQ.

(d) OBQA.          (e) PIQA.          (f) WinoGrande.

Figure 5: When applying the proposed layer-conditioned locating algorithm, the average loss of 64 samples from each dataset.

Figure 5 illustrates the average loss in model predictions when a single attention head is masked in the current layer, after previous layers have been masked, using the layer-conditioned locating algorithm. As can be seen, the proposed method can find some attention heads that have a much larger impact on model loss compared to masking a single attention head. This indicates that the proposed method effectively identifies a series of attention heads, termed knowledge circuits, that significantly influence the training loss of large language models.

## A.3 AVERAGE LOSS CURVES OF DIFFERENT LOCATING METHODS

Figure 6 shows the loss curves of different location methods across various datasets w.r.t layer of LLMs. It can be observed that, compared to random selection and simple greedy methods, our

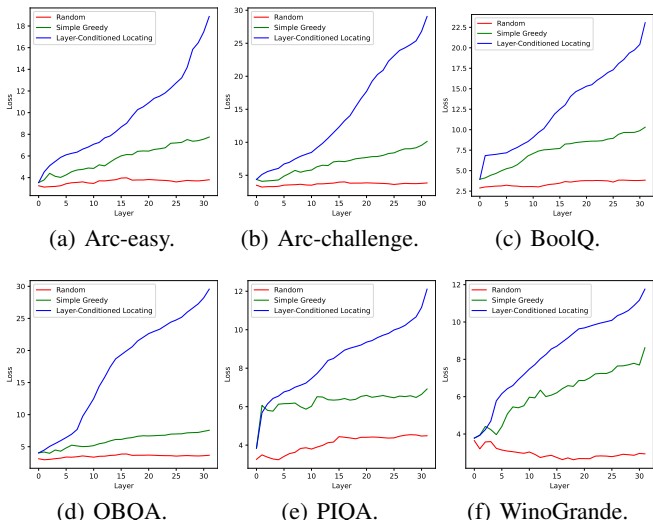

Figure 6: The average loss w.r.t. layer of LLMs on all datasets.

proposed layer-conditioned locating algorithm more accurately identifies the attention heads with the greatest impact on model training loss.

## A.4 AVERAGE LOSS CURVES ON ALL SAMPLES AND SELECTED SAMPLES OF EACH DATASET

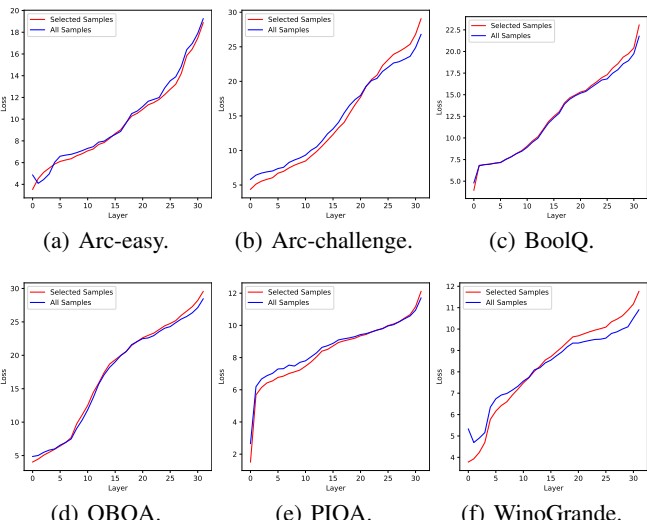

Figure 7: The average loss w.r.t. layer of LLMs on all samples of datasets.

To demonstrate that using samples with minimal loss effectively represents the model's ability to handle the corresponding task, we plotted the loss across layers for all samples and for those with minimal loss, as shown in Figure 7. As can be seen, the curves align closely, indicating that the loss variations in samples with the smallest loss reflect those of all samples. This suggests that using samples with the smallest loss to identify knowledge circuits is an efficient and effective approach.

## A.5 SIMILARITY OF KNOWLEDGE CIRCUITS

Figure 8 shows the similarity of knowledge circuits identified across different datasets on LLaMA3-8B. Most tasks exhibit very low similarity, while a few related tasks, such as ARC-e and ARC-c, have higher similarity. This highlights the potential of using knowledge circuits to analyze the behavior of similar tasks.

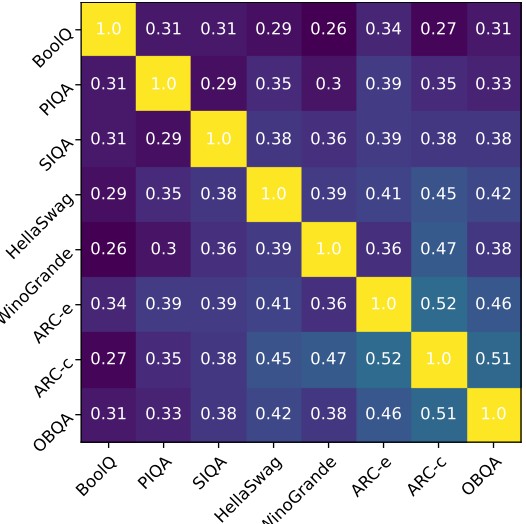

Figure 8: The similarity of selected heads between different tasks on LLaMA3-8B.

## A.6 PATTERN OF KNOWLEDGE CIRCUITS

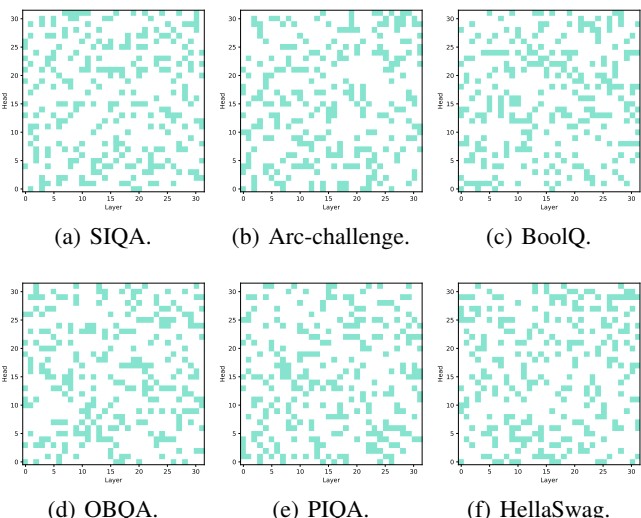

Figure 9: The located knowledge circuit of each dataset.

Figure 9 presents the patterns of knowledge circuits located by the layer-conditioned locating algorithm across different datasets. While some datasets exhibit similar patterns, the majority show significant variations. This motivates us to search for distinct knowledge circuits in different datasets.

## B  IMPLEMENTED DETAILS

Table 5 presents the learning rates of LoRA, HeadMap, and their combination across different NLU datasets. For HeadMap combined with the LoRA method, the former values in parentheses indicate the learning rate for HeadMap and the latter is for LoRA.

## C  ABLATION STUDY

**Transferability.**    In order to explore whether the knowledge circuit has transferability, we enhance the knowledge circuit identified in the WinoGrande dataset on different datasets. The results are pre-

Table 5: Learning rate of each baseline and each dataset.

| Method | CoLA | MRPC | RTE | SST-2 | WNLI |
|---|---|---|---|---|---|
| *LLaMA2-7B* | | | | | |
| LoRA | 2e-4 | 8e-4 | 8e-4 | 2e-4 | 1e-3 |
| HeadMap | 1e-3 | 8e-4 | 8e-4 | 1e-4 | 1e-3 |
| + LoRA (V, U, D) | (1e-4, 4e-4) | (1e-4, 8e-4) | (1e-3, 8e-4) | (1e-3, 2e-4) | (8e-4, 4e-4) |
| HeadMap$_{HS}$ | 8e-4 | 8e-4 | 1e-3 | 2e-4 | 8e-4 |
| + LoRA (V, U, D) | (8e-4, 4e-4) | (1e-4, 1e-3) | (1e-3, 8e-4) | (2e-4, 2e-4) | (2e-4, 1e-3) |
| *LLaMA3-8B* | | | | | |
| LoRA | 2e-4 | 8e-4 | 4e-4 | 2e-4 | 2e-4 |
| HeadMap | 4e-4 | 8e-4 | 8e-4 | 1e-4 | 1e-3 |
| + LoRA (V, U, D) | (1e-4, 4e-4) | (2e-4, 2e-4) | (1e-4, 4e-4) | (1e-4, 2e-4) | (8e-4, 8e-4) |
| HeadMap$_{HS}$ | 8e-4 | 1e-3 | 8e-4 | 2e-4 | 4e-4 |
| + LoRA (V, U, D) | (4e-4, 1e-4) | (1e-3, 2e-4) | (2e-4, 4e-4) | (4e-4, 1e-4) | (8e-4, 8e-4) |

Table 6: Accuracy (%) on eight commonsense reasoning datasets using LLaMA2-7B and LLaMA3-8B for different selection strategies. HeadMap is for outputs of MHA blocks. HeadMap$_{HS}$ is for hidden states.

| Method | BoolQ | PIQA | SIQA | HellaSwag | WinoGrande | ARC-e | ARC-c | OBQA | Avg. |
|---|---|---|---|---|---|---|---|---|---|
| *LLaMA2-7B* | | | | | | | | | |
| HeadMap$_{MHA}$ (Redundant) | 62.35 | 81.88 | 77.33 | 91.60 | 77.74 | 77.40 | 54.01 | 76.60 | 74.86 |
| HeadMap$_{MHA}$ (Random) | 62.63 | 82.48 | 78.25 | 91.88 | 78.14 | 78.07 | 54.78 | 75.00 | 75.15 |
| HeadMap$_{MHA}$ (Transfer from Winogrande) | 64.04 | 82.86 | 79.06 | 92.53 | 79.40 | 79.46 | 56.40 | 77.40 | 76.39 |
| HeadMap$_{MHA}$ | 63.94 | 83.19 | 79.07 | 92.74 | 79.40 | 80.30 | 57.00 | 78.60 | 76.78 |
| HeadMap$_{HS}$ (Redundant) | 63.03 | 82.37 | 76.92 | 91.37 | 78.30 | 77.99 | 53.42 | 75.80 | 74.90 |
| HeadMap$_{HS}$ (Random) | 63.27 | 83.57 | 78.30 | 92.51 | 76.64 | 78.79 | 53.41 | 75.40 | 75.24 |
| HeadMap$_{HS}$ (Transfer from Winogrande) | 64.53 | 82.97 | 78.92 | 92.58 | 78.37 | 79.76 | 54.95 | 77.60 | 76.21 |
| HeadMap$_{HS}$ | 63.73 | 83.30 | 79.73 | 92.63 | 78.37 | 80.30 | 55.38 | 78.20 | 76.46 |

Table 7: Time consumption of layer-conditioned locating algorithm on each dataset.

| | BoolQ | PIQA | SIQA | HellaSwag | WinoGrande | ARC-e | ARC-c | OBQA |
|---|---|---|---|---|---|---|---|---|
| Cost Time (min) | 3.58 | 22.13 | 8.62 | 22.2 | 6.53 | 16.68 | 17.30 | 11.10 |

sented in Table 6. It can be observed that compared to random selection, the transferred knowledge circuit demonstrates better performance. This suggests that some of the attention heads within the knowledge circuit contain general knowledge, not merely task-specific or dataset-specific knowledge. Hence, the knowledge circuits located by the layer-conditioned locating algorithm are capable of generalizing to other tasks or datasets.

**Rebundant.** We conduct additional ablation studies on redundant heads. Specifically, we randomly select 8 attention heads from redundant heads in each layer and enhance these attention heads. The results are shown in Table 6. It can be observed that compared to a fully random selection in each layer, the performance is worse when selecting and enhancing attention heads from redundant heads. This indicates that the assistance provided by these redundant heads for the model's adaptation to downstream tasks is more limited.

## D  TIME CONSUMPTION OF LAYER-CONDITIONED LOCATING

The time consumption of layer-conditioned locating is acceptable, as only a few samples require inference. This is further supported by the time required for knowledge circuit localization on each dataset, as shown in Table 7. Moreover, the localization procedure needs to be conducted only once per dataset, or even across multiple datasets, since experiments demonstrate that the located knowledge circuits are transferable.

