# OpenReview forum: "HeadMap: Locating and Enhancing Knowledge Circuits in LLMs"
_ICLR.cc/2025/Conference — ICLR 2025 Poster_

### Official Review · Reviewer_mNPV · 2024-10-30

**Soundness:** 3
**Presentation:** 3
**Contribution:** 2
**Rating:** 6
**Confidence:** 4

**Summary:**

This paper ‌explores‌ the intrinsic mechanisms underlying the attention heads of LLMs, and ‌proposes‌ an algorithm to detect the important heads that ‌play‌ a critical role in LLMs. Based on these observations, the authors further ‌propose‌ a new PEFT method that only fine-tunes these important heads, and ‌verify‌ its effectiveness via extensive experiments.

**Strengths:**

The overall writing is good. This paper is easy to follow.

The analysis about the  attention heads is comprehensive and reasonable.

The experimental results demonstrate the state-of-the-art performance, and the ablation study empirically proves the effectiveness of the proposed method.

**Weaknesses:**

The concept of the knowledge circuit is interesting, but the algorithm to find such a circuit is overly greedy. From my perspective, the selected heads are only layer-wise optimal. It seems this problem can be ‌solved‌ by dynamic programming to find a more optimal result, and ‌wouldn't‌ cost much. And then, the knowledge circuit can become a real "circuit".

The improvement of adopting HeadMap is not significant in Table 1, ‌especially‌ when using Llama3-8B as the LLM.

**Questions:**

Please see the weaknesses.

---

> ### Author Response · Authors · 2024-11-25
> **Response to Reviewer mNPV**
>
> Thank you for your constructive comments. Below we have made responses to your comments. If you have any further comment, please feel free to let us know and we are more than glad to discuss with you.
>
> > Q1. The concept of the knowledge circuit is interesting, but the algorithm to find such a circuit is overly greedy. From my perspective, the selected heads are only layer-wise optimal. It seems this problem can be ‌solved‌ by dynamic programming to find a more optimal result, and ‌wouldn't‌ cost much. And then, the knowledge circuit can become a real "circuit".
>
> Thank you for your suggestion. Our proposed method represents an initial exploration of identifying and applying knowledge circuits. Your suggestion to use dynamic programming for locating knowledge circuits is excellent, and we will explore it in our future work.
>
> > Q2. The improvement of adopting HeadMap is not significant in Table 1, ‌especially‌ when using Llama3-8B as the LLM.
>
> As shown in Table 1, the proposed method, HeadMap, **achieves comparable performance with significantly fewer trainable parameters** than LoRA and DoRA (i.e., less than 10% of trainable parameters). Combining HeadMap with LoRA achieves better performance with fewer parameters (only 2/3 of the trainable parameters) and shows improvements on most datasets. Therefore, the proposed method **not only improves the performance** but also **achieves better parameter efficiency**.

---

### Official Review · Reviewer_37tq · 2024-11-03

**Soundness:** 2
**Presentation:** 3
**Contribution:** 2
**Rating:** 5
**Confidence:** 4

**Summary:**

This paper introduces a novel method called HeadMap, aimed at identifying and enhancing knowledge circuits within large language models (LLMs). The authors propose a layer-conditioned locating algorithm that identifies critical attention heads layer by layer, which are essential for making accurate predictions on specific tasks. Through extensive experiments on various commonsense reasoning datasets, the authors demonstrate the effectiveness of the HeadMap method in improving model performance while maintaining parameter efficiency. Additionally, the paper explores the overlap of knowledge circuits across different datasets, revealing the varying knowledge requirements for different tasks.

**Strengths:**

1.	The proposed layer-conditioned locating algorithm and the HeadMap method provide a fresh perspective on understanding and optimizing LLMs, particularly in identifying and leveraging critical attention heads, showcasing innovation.
2.	HeadMap outperforms random selection and simple greedy methods, and the HeadMap method achieves comparable performance to baseline models while using fewer parameters

**Weaknesses:**

1.	While the paper shows that masking certain attention heads affects performance, it does not provide in-depth mechanistic insights into why these specific heads are critical.
2.	The layer-conditioned locating algorithm may introduce bias in selecting attention heads based on the specific datasets used. If the algorithm is overly tuned to the characteristics of these datasets, it may not generalize well to other tasks or datasets
3.	The experiments primarily focus on commonsense reasoning tasks. How does it perform in tasks like generative tasks, domain-specific applications?
4.	Lack of Ablation Studies on Redundant Heads

**Questions:**

Refer to Weaknesses.

---

> ### Author Response · Authors · 2024-11-25
> **Response to Reviewer 37tq (1/2)**
>
> Thank you for your constructive comments. Below we have made responses to your comments. If you have any further comment, please feel free to let us know and we are more than glad to discuss with you.
>
> > Q1. While the paper shows that masking certain attention heads affects performance, it does not provide in-depth mechanistic insights into why these specific heads are critical.
>
> According to previous studies [r1,r2,r3,r4], certain attention heads in the model store crucial knowledge for accurate task predictions and influence model predictions by injecting this knowledge into the residual stream. Hence, the focus of this paper is on how to identify and utilize the knowledge of those key attention heads for downstream tasks. Since downstream tasks are of different types, it is challenging to analyze why certain attention heads are critical for specific tasks. However, we consider this a valuable direction and plan to pursue it in future work.
>
> [r1] Knowledge Circuits in Pretrained Transformers. NeuriPS, 2024.
>
> [r2] Interpreting and Improving Large Language Models in Arithmetic Calculation. ICML, 2024.
>
> [r3] Cutting Off the Head Ends the Conflict: A Mechanism for Interpreting and Mitigating Knowledge Conflicts in Language Models. ACL, 2024.
>
> [r4] Locating and Editing Factual Associations in GPT. NeuriPS, 2022.
>
>
> > Q2. The layer-conditioned locating algorithm may introduce bias in selecting attention heads based on the specific datasets used. If the algorithm is overly tuned to the characteristics of these datasets, it may not generalize well to other tasks or datasets.
>
> Thanks for your valuable suggestions. To explore whether the layer-conditioned locating algorithm introduces the task-related bias into the knowledge circuit, we evaluated the knowledge circuit identified in the WinoGrande dataset on different datasets. The results are presented in the following table. It can be observed that compared to random selection, the transferred knowledge circuit demonstrates better performance. This suggests that **some of the attention heads within the knowledge circuit contain general knowledge**, not merely task-specific or dataset-specific knowledge. Hence, the knowledge circuits located by the layer-conditioned locating algorithm are capable of generalizing to other tasks or datasets.
>
> |  | BoolQ | PIQA | SIQA | HellaSwag | WinoGrande | ARC-e | ARC-c | OBQA | Avg |
> | ----- | ------ | ------ | ------ | ------ | ------ | ------ | ------ | ------ | ------ |
> | HeadMap$_{\text{MHA}}$ (Random) | 62.63 | 82.48 | 78.25 | 91.88 | 78.14 | 78.07 | 54.78 | 75.00 | 75.1
> | HeadMap$_{\text{MHA}}$ (Transfer) | 64.04 | 82.86 | 79.06 | 92.53 | 79.40 | 79.46 | 56.40 | 77.40 | 76.39 |
> HeadMap$_{\text{HS}}$ (Random) | 63.27 | 83.57 | 78.30 | 92.51 | 76.64 | 78.79 | 53.41 | 75.40 | 75.24 |
> |HeadMap$_{\text{HS}}$ (Transfer) | 64.53 | 82.97 | 78.92 | 92.58 | 78.37 | 79.76 | 54.95 | 77.60 | 76.21 |
>
>
>
> > Q3. The experiments primarily focus on commonsense reasoning tasks. How does it perform in tasks like generative tasks, domain-specific applications?
>
> In addition to the commonsense reasoning task, we also evaluated our proposed method on natural language understanding tasks. As shown in Table 2, our method exhibits significant improvement compared to LoRA.
>
> Moreover, following your suggestion, we evaluated our method on the MathQA dataset [r5]. The MathQA dataset contains 37k math word problems. The results are shown in the following table. As can be seen, the proposed method, HeadMap, **achieves comparable performance with significantly fewer trainable parameters** than LoRA (requiring less than 10% of trainable parameters). Combining HeadMap with LoRA achieves better performance with fewer parameters (only about 2/3 of the trainable parameters) and shows improvements on this dataset. Therefore, the proposed method **not only improves performance** but also **achieves better parameter efficiency** on the MathQA dataset.
>
>
> | Method | # Params (%) | MathQA |
> | ------ | ------ | ------ |
> | LoRA   | 0.94 | 40.34 |
> |HeadMap$_{\text{MHA}}$ | 0.08 | 40.10 |
> |&ensp;  + LoRA (V, U, D) | 0.66 | 42.14 |
> |HeadMap$_{\text{HS}}$ |  0.08 | 39.63 |
> |&ensp;  + LoRA (V, U, D) | 0.66 | 41.88 |
>
> [r5] MathQA: Towards Interpretable Math Word Problem Solving with Operation-Based Formalisms. NAACL, 2019.

---

> ### Author Response · Authors · 2024-11-25
> **Response to Reviewer 37tq (2/2)**
>
> > Q4. Lack of Ablation Studies on Redundant Heads
>
> Thanks for your valuable suggestions. We conduct additional ablation studies on redundant heads. Specifically, we randomly select 8 attention heads from redundant heads (i.e. the attention heads that are not in located knowledge circuit) in each layer and enhance these attention heads. The results are shown in the following table. It can be observed that compared to a fully random selection in each layer, the performance is worse when selecting and enhancing attention heads from redundant heads. This indicates that the assistance provided by these redundant heads for the model's adaptation to downstream tasks is more limited. As suggested, the ablation study of redundant attention heads has been added in Appendix C of the revised manuscript.
>
> |  | BoolQ | PIQA | SIQA | HellaSwag | WinoGrande | ARC-e | ARC-c | OBQA | Avg |
> | ----- | ------ | ------ | ------ | ------ | ------ | ------ | ------ | ------ | ------ |
> | HeadMap$_{\text{MHA}}$ (Redundant) | 62.35 | 81.88 | 77.33 | 91.60 | 77.74 | 77.40 | 54.01 | 76.60 | 74.86 |
> | HeadMap$_{\text{MHA}}$ (Random) | 62.63 | 82.48 | 78.25 | 91.88 | 78.14 | 78.07 | 54.78 | 75.00 | 75.10 |
> | HeadMap$_{\text{HS}}$ (Redundant) | 63.03 | 82.37 | 76.92 | 91.37 | 78.30 | 77.99 | 53.42 | 75.80 | 74.90 |
> HeadMap$_{\text{HS}}$ (Random) | 63.27 | 83.57 | 78.30 | 92.51 | 76.64 | 78.79 | 53.41 | 75.40 | 75.24 |

---

> > ### Comment · Reviewer_37tq · 2024-11-27
> >
> > Thank you for the clarification. I have no further question, however I would like to keep my score.

---

### Official Review · Reviewer_pHFG · 2024-11-04

**Soundness:** 3
**Presentation:** 3
**Contribution:** 2
**Rating:** 5
**Confidence:** 4

**Summary:**

This paper attempts to explore the intrinsic mechanism of LLMs with the concept of knowledge circuits. Specifically, the termed knowledge circuits refer to the attention heads that are more important to specific tasks in all attention heads across all transformer layers. To locate the knowledge circuits, the paper explores direct strategy, simple greedy strategy and layer-conditioned strategy. The underlying assumption is that if the attention head is important, masking this attention head would cause significant increase of the task loss. After locating knowledge circuits, the paper fine-tunes the attention heads therein with LoRA-like Map modules. Experimental results evaluate the effectiveness of this design.

**Strengths:**

1. Knowledge circuit is an interesting concept that can help understand the mechanism of LLM performance on specific tasks.
2. The paper proposes three strategies to locate the knowledge circuits, and two strategies to fine-tune the model with knowledge circuits, which can be inspirable for future works on knowledge circuits.
3. Experimental results demonstrate the effectiveness of the proposed model.

**Weaknesses:**

1. The use of locating knowledge circuits is somewhat straightforward, and the concept of knowledge circuit was proposed be existing studies [1]. Actually, the reference [1] is missing in this paper, and it requires to make discussion with [1].
2. The proposed method may have less efficiency and less generality. The knowledge circuit has to be located specifically for specific tasks, and then be adopted to enhance the model on the target task. The method requires to detect the 64 samples with lowest losses, and then conduct layer-conditioned locating by masking each attention head in each layer one-by-one. This procedure would have high time-consume. Besides, the knowledge circuit may hardly be transferred on different tasks. This would cause the limitation of the proposed method in application. The generality is also concerned whether the method can be used for different tasks.
3. The selected samples with lowest losses may also be dubious. The 64 samples with lowest losses may be the 64 most easy samples for each task. Therefore, the located knowledge circuit may not be optimal.
4. The performance improvements are somewhat marginal. This also affects the contribution of the proposed method.

[1] Knowledge Circuits in Pretrained Transformers. NeurIPS 24.

**Questions:**

1. In Figure 1 (a) and (b), it has different value ranges of attention visualization between the results on SIQA and HellaSwag. What may be the reason of that? Why there is only a light point in Figure 1(d) on Layer-0 and Head-25?
2. What are the impacts of other attention heads beside the knowledge circuits? Can these attention heads be ablated to further enhance the accuracy and efficiency?

---

> ### Author Response · Authors · 2024-11-25
> **Response to Reviewer pHFG (1/3)**
>
> Thank you for your constructive comments. Below we have made responses to your comments. If you have any further comment, please feel free to let us know and we are more than glad to discuss with you.
>
> > Q1. The use of locating knowledge circuits is somewhat straightforward, and the concept of knowledge circuit was proposed be existing studies [r1]. Actually, the reference [r1] is missing in this paper, and it requires to make discussion with [r1].
> >
> > [r1] Knowledge Circuits in Pretrained Transformers. NeurIPS 24.
>
> Thanks for your valuable suggestions. In our revised manuscript, we have included discussions with [r1].
>
> * Discussions on [r1]:
>     1. **The focus of the tasks differs.** Reference [r1] primarily examines the components of the Transformer that play a crucial role in factual recall tasks. In contrast, this paper investigates the knowledge circuits across a wide range of tasks and datasets, not limited to factual recall tasks.
>     2. **The components studied differ.** Reference [r1] analyzes both the MLP components and attention heads within the Transformer architecture, whereas this paper primarily conducts experiments on the attention heads.
>     3. **The methods for identifying knowledge circuits differ.** In [r1], knowledge circuits are located by removing task-irrelevant components, whereas this paper identifies them by progressively removing important attention heads.
>     4. **The target scenarios differ.** In [r1], the discovered knowledge circuits are used to analyze and interpret model behavior and knowledge editing techniques. In contrast, this paper suggests that strengthening knowledge circuits can enhance model performance on downstream tasks.
>
> In summary, based on the above discussions, our work differs with reference [r1].
>
>
> > Q2. The method requires to detect the 64 samples with lowest losses, and then conduct layer-conditioned locating by masking each attention head in each layer one-by-one. This procedure would have high time-consume.
>
> We think that **the time consumption of layer-conditioned locating is acceptable.**
>
> 1. We only need to **perform inference on a small number of samples**, thus introduce acceptable time consumption. To see that, we record in the following table the time required for knowledge circuit localization on each dataset.
> 1. We only need to **perform the locating procedure once for each dataset or even multiple datasets**, since experiments shown in our response to Q3 demonstrate that the located knowledge circuits are transferable.
>
>
> | | BoolQ | PIQA | SIQA | HellaSwag | WinoGrande | ARC-e | ARC-c | OBQA |
> | ------ | ------ | ------ | ------ | ------ | ------ | ------ | ------ | ------ |
> | Cost Time (min) | 3.58 | 22.13 | 8.62 | 22.2 | 6.53 | 16.68 | 17.30 | 11.10 |
> <!-- | Cost Time (s) | 214.53 | 1327.87 | 516.97 | 1331.74 | 392.04 | 1001.06 | 1038.21 | 665.72 | -->
>
>
> > Q3. Besides, the knowledge circuit may hardly be transferred on different tasks.
>
> We find that the knowledge circuit identified on a specific task or dataset **exhibits transferability**. Specifically, we apply the knowledge circuit located from the WinoGrande dataset to fine-tune models on different datasets using the HeadMap method, with the results shown in the following table. As can be seen, the knowledge circuit identified by the proposed HeadMap method outperforms randomly selected attention heads. This indicates that the knowledge circuit identified by our method is transferable and demonstrates robustness across tasks and datasets. This enhances the practical value of the proposed method.
>
> As suggested, the experiment of transferability of knowledge circuits has been added in Appendix C of the revised manuscript.
>
> |  | BoolQ | PIQA | SIQA | HellaSwag | WinoGrande | ARC-e | ARC-c | OBQA | Avg |
> | ----- | ------ | ------ | ------ | ------ | ------ | ------ | ------ | ------ | ------ |
> | HeadMap$_{\text{MHA}}$ (Random) | 62.63 | 82.48 | 78.25 | 91.88 | 78.14 | 78.07 | 54.78 | 75.00 | 75.1
> | HeadMap$_{\text{MHA}}$ (Transfer) | 64.04 | 82.86 | 79.06 | 92.53 | 79.40 | 79.46 | 56.40 | 77.40 | 76.39 |
> HeadMap$_{\text{HS}}$ (Random) | 63.27 | 83.57 | 78.30 | 92.51 | 76.64 | 78.79 | 53.41 | 75.40 | 75.24 |
> |HeadMap$_{\text{HS}}$ (Transfer) | 64.53 | 82.97 | 78.92 | 92.58 | 78.37 | 79.76 | 54.95 | 77.60 | 76.21 |

---

> ### Author Response · Authors · 2024-11-25
> **Response to Reviewer pHFG (2/3)**
>
> > Q4. The generality is also concerned whether the method can be used for different tasks.
>
> We have evaluated our proposed method on commonsense reasoning tasks and natural language understanding tasks. As shown in Table 1 and 2, our method exhibits significant improvement compared to LoRA.
>
> Moreover, we evaluated our method on the MathQA dataset [r2]. The MathQA dataset contains 37k math word problems. The results are shown in the following table. As can be seen, the proposed method, HeadMap, **achieves comparable performance with significantly fewer trainable parameters** than LoRA (requiring less than 10% of trainable parameters). Combining HeadMap with LoRA achieves better performance with fewer parameters (only about 2/3 of the trainable parameters) and shows improvements on this dataset. Therefore, the proposed method **not only improves performance** but also **achieves better parameter efficiency**. Experiments on diverse datasets and tasks validated the effectiveness and generalizability of the proposed method.
>
> | Method | # Params (%) | MathQA |
> | ------ | ------ | ------ |
> | LoRA   | 0.94 | 40.34 |
> |HeadMap$_{\text{MHA}}$ | 0.08 | 40.10 |
> |&ensp;  + LoRA (V, U, D) | 0.66 | 42.14 |
> |HeadMap$_{\text{HS}}$ |  0.08 | 39.63 |
> |&ensp;  + LoRA (V, U, D) | 0.66 | 41.88 |
>
> [r2] MathQA: Towards Interpretable Math Word Problem Solving with Operation-Based Formalisms. NAACL, 2019.
>
> > Q5. The selected samples with lowest losses may also be dubious. The 64 samples with lowest losses may be the 64 most easy samples for each task. Therefore, the located knowledge circuit may not be optimal.
>
> We think that samples with the smallest loss do not necessarily mean they are the simplest. Instead, they correspond to samples, in which the model are the most confident, and hence those samples could reflect the knowledge embedded in the model for handling corresponding tasks. Moreover, this method of selecting samples has also been used in previous works [r3,r4,r5,r6].
>
> To further support our viewpoint, we plotted the curve of average loss across all samples in the dataset w.r.t. layers. A comparison with the average loss of the selected samples is shown in Appendix A.4. As can be seen, the two curves align closely, indicating that the loss variations in samples with the smallest loss reflect those of all samples. This suggests that using samples with the smallest loss to identify knowledge circuits is an efficient and effective approach.
>
> [r3] Interpreting and Improving Large Language Models in Arithmetic Calculation. ICML 2025.
>
> [r4] Interpretable catastrophic forgetting of large language model fine-tuning via instruction vector. arXiv preprint arXiv:2406.12227, 2024.
>
> [r5] Function Vectors in Large Language Models. ICLR, 2024.
>
> [r6] Knowledge Circuits in Pretrained Transformers. NeurIPS 24.
>
> > Q6. The performance improvements are somewhat marginal. This also affects the contribution of the proposed method.
>
> As shown in Tables 1 and 2, the proposed HeadMap method achieves **comparable performance with significantly fewer trainable parameters** than LoRA and DoRA (requiring less than 10% of trainable parameters). Combining HeadMap with LoRA achieves better performance with fewer parameters (only 2/3 of the trainable parameters) and shows improvements on most datasets. Therefore, the proposed method **not only improves performance** but also **achieves better parameter efficiency**.
>
>
> > Q7. In Figure 1 (a) and (b), it has different value ranges of attention visualization between the results on SIQA and HellaSwag. What may be the reason of that?
>
> * The difference value ranges between Figures 1(a) and 1(b) suggests that masking knowledge circuits located by the layer-conditioned locating algorithm has a more significant impact on the prediction loss than masking individual attention heads.
> * For Figures 1(a) and 1(d), different value ranges of attention visualization indicate that different attention heads have varying importance for different tasks (e.g., SIQA and HellaSwag), influenced by factors such as task difficulty and required knowledge. Additionally, for a given task, the impact of masking a single attention head on model inference can also vary.
>
> > Q8. Why there is only a light point in Figure 1(d) on Layer-0 and Head-25?
>
> Regarding the influential attention head in Figure 1(d) (i.e., a light point in Figure 1(d)), we consider Head-25 in Layer-0 to play a crucial role in model predictions. However, since its knowledge is integrated into the residual stream in later layers, it already aids the model in making accurate predictions, so masking other attention heads does not have a substantial effect.

---

> ### Author Response · Authors · 2024-11-25
> **Response to Reviewer pHFG (3/3)**
>
> > Q9. What are the impacts of other attention heads beside the knowledge circuits? Can these attention heads be ablated to further enhance the accuracy and efficiency?
>
> Thanks for your valuable suggestions.
>
> * We think that other attention heads beside the knowledge circuits may provide the model with some task-irrelevant but more fundamental knowledge, such as grammatical knowledge. To determine whether other attention heads are task-independent, we conduct additional ablation studies on other attention heads that are not in the knowledge circuits. Specifically, we randomly select 8 attention heads from other heads beside the knowledge circuits in each layer and enhance those attention heads. The results are shown in the following table. It can be observed that compared to a fully random selection in each layer, the performance is worse when selecting from other attention heads beside the knowledge circuits. This indicates that the assistance provided by those attention heads for the model's adaptation to downstream tasks is more limited. We have included this ablation study in Appendix C of the revised manuscript.
>
> |  | BoolQ | PIQA | SIQA | HellaSwag | WinoGrande | ARC-e | ARC-c | OBQA | Avg |
> | ----- | ------ | ------ | ------ | ------ | ------ | ------ | ------ | ------ | ------ |
> | HeadMap$_{\text{MHA}}$ (Redundant) | 62.35 | 81.88 | 77.33 | 91.60 | 77.74 | 77.40 | 54.01 | 76.60 | 74.86 |
> | HeadMap$_{\text{MHA}}$ (Random) | 62.63 | 82.48 | 78.25 | 91.88 | 78.14 | 78.07 | 54.78 | 75.00 | 75.1 |
> | HeadMap$_{\text{HS}}$ (Redundant) | 63.03 | 82.37 | 76.92 | 91.37 | 78.30 | 77.99 | 53.42 | 75.80 | 74.90 |
> HeadMap$_{\text{HS}}$ (Random) | 63.27 | 83.57 | 78.30 | 92.51 | 76.64 | 78.79 | 53.41 | 75.40 | 75.24 |
>
> * We found that simply removing other attention heads beside the knowledge circuits is challenging.
>     * First, it is unreasonable to remove them, as identifying more important attention heads does not imply that the others are useless. For example, some attention heads store grammatical knowledge, which, although not critical for the task, ensures the output's coherence.
>     * Second, we have conducted experiments which show that directly removing other attention heads beside the knowledge circuits can lead to training issues like non-convergence or NaN. Such an approach requires more careful design of training strategies and parameter initialization.

---

> > ### Comment · Reviewer_pHFG · 2024-11-27
> >
> > Thanks for the authors' responses. The additional results improve the quality of this paper. However, I still concern the novelty, and the in-depth mechanistic insights are insufficient. Therefore, I would like to improve score but retain a negative score.

---

> > > ### Author Response · Authors · 2024-11-29
> > >
> > > Thank you for your valuable suggestions, which have helped us improve the quality of the paper. Below are our responses to your concerns regarding novelty and the in-depth mechanistic insights. If you have any further comment, please feel free to let us know and we are more than glad to discuss with you.
> > >
> > > > Q1. However, I still concern the novelty, and the in-depth mechanistic insights are insufficient.
> > >
> > > * Novelty: As mentioned in the paper, unlike previous works [r1,r2,r3,r4], our proposed method, the layer-conditioned locating algorithm, is a general approach. Previous methods for identifying key attention heads and MLP layers were designed based on constructed sentence formats, often requiring detailed modifications within these sentences. While this can be more precise, it is difficult to generalize to a broader range of tasks. The work most similar to ours is [r5], but his approach is opposite to ours. We determine the most crucial set of attention heads on model predictions by masking layer by layer, whereas [r5] retains the most important attention heads by gradually removing the least crucial ones. Additionally, based on the discovered knowledge circuits, we designed a parameter-efficient fine-tuning method that enhances the knowledge in these circuits within the residual stream to improve the model's adaptability to downstream tasks. This significantly differs from previous work. Therefore, we believe this article demonstrates significant novelty.
> > >
> > > * In-depth mechanistic insights: In this paper, due to the diversity of datasets and tasks we explore, it is challenging to determine how knowledge circuits work for constructing sentences and for a certain task, as done in previous works [r1,r2,r3,r4,r5]. Moreover, based on our experiments, we have made the following findings:
> > >     * Transferability of Knowledge circuits: As shown in Table 6 of Appendix C, by enhancing the knowledge circuits located in WinoGrande across different tasks and datasets, we observe better performance compared to randomly selected heads, indicating that knowledge circuits have transferability and some attention heads play similarly important roles across tasks.
> > >     * Effective Localization with Few Samples: A small number of correctly predicted samples can effectively identify important knowledge circuits of large language models for a larger dataset. Figure 7 in Appendix A.4 shows that the layer-conditioned locating algorithm, using 64 samples with the lowest loss, locates the knowledge circuit that is also crucial for other unseen samples in the dataset.
> > >     * Limited Impact of Many Attention Heads: Many attention heads have minimal impact on task-specific predictions. As shown in Figures 1 and 6, we find that masking the same number of attention heads results in very small changes in the prediction loss.
> > >
> > > [r1] Locating and Editing Factual Associations in GPT. NeurIPS 2022.
> > >
> > > [r2] Interpretability in the Wild: a Circuit for Indirect Object Identification in GPT-2 Small. ICLR 2023.
> > >
> > > [r3] Cutting Off the Head Ends the Conflict: A Mechanism for Interpreting and Mitigating Knowledge Conflicts in Language Models. ACL 2024.
> > >
> > > [r4] Interpreting and Improving Large Language Models in Arithmetic Calculation. ICML 2024.
> > >
> > > [r5] Knowledge Circuits in Pretrained Transformers. NeurIPS 2024.

---

> > > > ### Comment · Reviewer_pHFG · 2024-12-03
> > > >
> > > > Thanks for the authors' responses. I have no other questions.

---

### Official Review · Reviewer_Kvw1 · 2024-11-04

**Soundness:** 3
**Presentation:** 3
**Contribution:** 3
**Rating:** 5
**Confidence:** 5

**Summary:**

This paper represents another method of task-specific training and is one of the parameter-efficient fine-tuning approaches, referred to as HeadMap. The authors discovered that for certain tasks, specific attention heads are particularly influential; masking these heads significantly decreases performance. They propose a layer-conditioned locating algorithm to identify knowledge circuits in LLMs that greatly impact predictive accuracy. Based on this, they suggest training focused on these knowledge circuits, where only a small number of parameters are updated. The results are complementary to those of LoRA, and together they yield improved outcomes.

**Strengths:**

It complements LoRA-type fine-tuning with utilizing significantly fewer parameters. When combined, they enhance accuracy for specific tasks and could be a valuable method for various applications.

**Weaknesses:**

Compared to LoRA, this approach is more challenging to utilize in practice.

**Questions:**

Is the code available? How can we verify the results?

---

> ### Author Response · Authors · 2024-11-25
> **Response to Reviewer Kvw1**
>
> Thank you for your constructive comments. Below we have made responses to your comments. If you have any further comment, please feel free to let us know and we are more than glad to discuss with you.
>
> > Q1. Compared to LoRA, this approach is more challenging to utilize in practice.
>
> The proposed methods, layer-conditioned locating algorithm and HeadMap, are **as easy as LoRA** to be utilized in parctice.
>
> First, the layer-conditioned locating algorithm enables the identification of important attention heads (i.e. knowledge circuit) for a dataset or specific tasks **using only a small number of samples**. Due to the small sample size, the runtime of the algorithm is acceptable compared to full-scale training.
>
> Second, we find that the knowledge circuit identified on a specific task or dataset **exhibits transferability**. Specifically, we apply the knowledge circuit located from the WinoGrande dataset to fine-tune models on different datasets using the HeadMap method, with the results shown in the following table. As can be seen, enhancing the transferred knowledge circuit outperforms enhancing randomly selected attention heads. This indicates that the knowledge circuits identified by the layer-conditioned locating algorithm are transferable and demonstrate robustness across tasks and datasets. This enhances the practical value of the proposed method.
>
>
> |  | BoolQ | PIQA | SIQA | HellaSwag | WinoGrande | ARC-e | ARC-c | OBQA | Avg |
> | ----- | ------ | ------ | ------ | ------ | ------ | ------ | ------ | ------ | ------ |
> | HeadMap$_{\text{MHA}}$ (Random) | 62.63 | 82.48 | 78.25 | 91.88 | 78.14 | 78.07 | 54.78 | 75.00 | 75.10
> | HeadMap$_{\text{MHA}}$ (Transfer) | 64.04 | 82.86 | 79.06 | 92.53 | 79.40 | 79.46 | 56.40 | 77.40 | 76.39 |
> HeadMap$_{\text{HS}}$ (Random) | 63.27 | 83.57 | 78.30 | 92.51 | 76.64 | 78.79 | 53.41 | 75.40 | 75.24 |
> |HeadMap$_{\text{HS}}$ (Transfer) | 64.53 | 82.97 | 78.92 | 92.58 | 78.37 | 79.76 | 54.95 | 77.60 | 76.21 |
>
>
> Third, as shown in Tables 1 and 2, the proposed method achieves higher parameter efficiency and comparable performance compared to LoRA and combining HeadMap with LoRA yields better results using fewer parameters. This demonstrates the practical value of the proposed method.
>
> > Q2. Is the code available? How can we verify the results?
>
> Our code is provided in the supplementary materials. By following the procedure outlined in the paper, the results reported in the paper could be reproduced.

---

### Meta-Review · Area_Chair_KDdk · 2024-12-20

**Metareview:**

This paper proposes a new parameter efficient fine-tuning technique that involves identifying knowledge circuits (attention heads that, when masked, lead to drop in performance) in transformer models, and then selectively fine-tuning these. The method achieves comparable performance with LoRA and Dora, but utilizes lesser number of parameters. Generally, reviewers find this contribution valuable. The paper should include more details about efficiency of the approach, especially about the added overhead from the required full fine-tuning steps to identify the knowledge circuits.

**Additional Comments On Reviewer Discussion:**

Reviewer pHFG raised the issue of marginal improvement over baselines. The authors address this point adequately. Multiple reviewers (Kvw1, pHFG) point out that the proposed approach is more difficult to use than LoRA. Although the authors stress that the number of parameters updated in their method are lower, their approach needs some number of full fine-tuning steps that may be quite limiting (e.g. if the available hardware cannot support full fine-tuning).
The experiment with redundant heads (as requested by reviewer pHFG) shows that there is negligible difference between masking redundant and random heads, casting some doubt over how important the selection of the heads is.

---

### Decision · Program_Chairs · 2025-01-22

Accept (Poster)